# Genome-Wide Association Analysis of Fruit Shape-Related Traits in *Areca catechu*

**DOI:** 10.3390/ijms24054686

**Published:** 2023-02-28

**Authors:** Hao Ding, Guangzhen Zhou, Long Zhao, Xinyu Li, Yicheng Wang, Chengcai Xia, Zhiqiang Xia, Yinglang Wan

**Affiliations:** Hainan Key Laboratory for Sustainable Utilization of Tropical Bioresources, College of Tropical Crops, Hainan University, Haikou 570228, China

**Keywords:** *Areca catechu* L., fruit shape, single-nucleotide polymorphism (SNP), genome-wide association analysis (GWAS), candidate gene

## Abstract

The areca palm (*Areca catechu* L.) is one of the most economically important palm trees in tropical areas. To inform areca breeding programs, it is critical to characterize the genetic bases of the mechanisms that regulate areca fruit shape and to identify candidate genes related to fruit-shape traits. However, few previous studies have mined candidate genes associated with areca fruit shape. Here, the fruits produced by 137 areca germplasms were divided into three categories (spherical, oval, and columnar) based on the fruit shape index. A total of 45,094 high-quality single-nucleotide polymorphisms (SNPs) were identified across the 137 areca cultivars. Phylogenetic analysis clustered the areca cultivars into four subgroups. A genome-wide association study that used a mixed linear model identified the 200 loci that were the most significantly associated with fruit-shape traits in the germplasms. In addition, 86 candidate genes associated with areca fruit-shape traits were further mined. Among the proteins encoded by these candidate genes were UDP-glucosyltransferase 85A2, the ABA-responsive element binding factor GBF4, E3 ubiquitin-protein ligase SIAH1, and LRR receptor-like serine/threonine-protein kinase ERECTA. Quantitative real-time polymerase chain reaction (qRT-PCR) analysis showed that the gene that encoded UDP-glycosyltransferase, *UGT85A2*, was significantly upregulated in columnar fruits as compared to spherical and oval fruits. The identification of molecular markers that are closely related to fruit-shape traits not only provides genetic data for areca breeding, but it also provides new insights into the shape formation mechanisms of drupes.

## 1. Introduction

Fruits, generally developed from the ovary of a plant, protect the seed for propagation and reproduction during plant growth and development [1,2]. Fruit is also an economically important organ for numerous crops. As one of the characteristics of fruit appearance, shape directly affects the commercial value and economic benefits of fruit [3,4]. Therefore, fruit shape is an influential selection feature in a variety of breeding programs [5]. Hormonal regulation, varietal characteristics, plant nutrient levels, and the features of the cultivation environment (such as temperature, light, and moisture) are the main factors that control fruit shape [6,7].

Fruit shape is a complex trait that is genetically controlled by multiple genes in different ways [8]. Recent extensive genetic studies of fruit shapes have resulted in a series of advances [3,9,10]. Tomato fruit shape has received particularly thorough research attention, and several genes have been shown to participate in regulating fruit shape [11,12,13,14]. For example, *SUN*, a member of the IQ67-domain (IQD) protein family that encodes calmodulin binding proteins, is highly expressed in tomato ovaries and young fruits; *SUN* expression not only affects the morphology of the vegetative organs, but it also leads to the elongation of the flower organs and fruits during peak cell division after pollination [11]. Transposon-mediated doubling events led to a significant increase in the *SUN* expression level and promoted fruit elongation [11,15]. *OVATE*, located on tomato chromosome 2, is one of the main members of the OVATE family, which inhibits fruit elongation [16,17]. The early appearance of a stop codon in the OVATE protein sequence, caused by a single base mutation, can alter the shape of the tomato fruit from round to pear-like. Previous studies have shown that both the *SOV1* locus on chromosome 10 and the *SOV2* locus on chromosome 11 can promote tomato elongation by inhibiting *OVATE* expression [16,17]. Furthermore, *FASCIATED* (*FAS*) and *LOCULE NUMBER* (*LC*) affect fruit shape by changing the locule number in tomatoes [18,19,20]. The previous study showed that MuMADS1 could interact with the OVATE family protein MaOFP1 and regulate fruit quality in a tomato ovate mutant [21]. In apples, the ectopic expression of *Malus domestica PISTILLATA* (*MdPI*), the ortholog of the floral organ identity gene *PISTILLATA* (*PI*), regulates apple fruit tissue growth and shape [22].

With the rapid development of high-throughput genotyping, genome-wide association analysis (GWAS) has recently become a powerful approach for explorations of the genetic mechanisms that underlie fruit-shape traits [23,24,25]. A previous GWAS identified a 1.7 Mb inversion on peach chromosome 6 that was associated with peach fruit shape [25]. Also near this region is *PpOFP1*, which inhibits vertical peach growth by interacting with an activated factor of fruit elongation, *PpTRM17*, resulting in flat-shaped fruits [25]. In addition, grape fruit-shape-related genes were obtained by GWAS analysis based on the whole-genome resequencing of 472 grape germplasm resources [26]. This study identified a SNP site (Chr7.20085154; mutation of “C” to “T”) on grape chromosome 7 associated with fruit shape regulation; this site was associated with a gene that encodes the serine/threonine protein kinase SRK2A [26,27]. Fruit-shape regulation mechanisms have been analyzed using GWAS in a variety of other fruits, including melon and watermelon [10,28]. These previous studies provide a framework that we can use to explore the mechanisms that regulate areca fruit shape.

*A. catechu*, a perennial casuarina in the palm family (*Arecaceae*), is endemic to Malaysia. The areca palm is one of the most economically important trees in the tropics due to its high medicinal and edible value [29,30]. Areca nuts, which are dried maturate areca fruit, are also a popular food, and more than 400 million people habitually eat areca nuts in China, South Asia, East Africa, and other countries and regions [29,31]. As Chinese consumers have a strong preference for oval areca nuts, it is important to clarify the molecular mechanisms that regulate areca fruit shape for use by the areca industry. Recently, substantial progress has been made in the understanding of fruit-shape regulation mechanisms in tomato, a berry, but little is known about the shape-forming mechanisms of areca fruit, a drupe. In this study, we performed the first GWAS of areca nuts to identify the loci associated with fruit-shaped regulation and to highlight useful molecular markers. We also explored the genetic structure and genetic diversity of areca palm populations.

## 2. Results

### 2.1. Genotype Analysis of A. catechu Populations

After the filtering of data, a total of 714.30 Gb of data were obtained through the sequencing of 137 areca genomic DNA sequences. Then, after filtering all the SNPs and indels (max missing > 0.5, minor allele frequency, MAF > 0.05 and Hardy–Weinberg equilibrium *p*-value > 0.001), we obtained 45,094 high-quality SNPs and 2347 indels across all 16 areca chromosomes (Figure 1). The numbers of variants per chromosome ranged from 1591 (chromosome 7) to 3992 (chromosome 10). These variants covered a total physical distance of ~2.45 Gb, with the shortest distance on chromosome 7 (92.0 Mb) and the longest distance on chromosome 15 (204.2 Mb). The genome-wide average variant density was 51.9 kb per variant and varied from 46.2 kb (chromosome 14) to 60.9 kb (chromosome 2) (Figure 1; Appendix A).

We detected the following two types of SNP markers in the areca genome: transitions (ts) and transversions (tv). ‘GA’-type mutations were the most common transitions (5927) and ‘TA’-type mutations were the most common transversions (3697). Across the 137 individuals tested, the average value of ts/tv was 2.56, the maximum ts/tv for a single sample was 2.82, and the minimum ts/tv for a single sample was 2.00 (Appendix A). Distribution analysis of the high-quality SNPs showed that 37,003 (65.8%) were located in the intergenic regions, 483 (0.9%) were located in the untranscribed regions, 8984 (15.98%) were located in introns, and only 1219 SNPs (2.17%) were located in coding regions (Figure 2; Appendix A). Notably, 340 SNPs (28.6%) in the coding regions produced silent mutations, and 793 SNPs (66.8%) produced missense mutations (Figure 2).

### 2.2. Genetic Diversity Revealed by SNP Markers

Across the 137 materials tested, fruit shape was roughly divided into spherical, oval, and columnar based on the fruit shape index. PCA and kindship analysis, performed using all high-quality SNPs and indel data, identified an obvious population structure that separated the Hainanese and Vietnamese populations (Appendix A). Our neighbor-joining phylogenetic tree clustered the areca germplasms with different fruit-shaped traits into four population subgroups. Areca trees with oval fruit were mainly clustered into subgroup 1, those with spherical fruit were mainly clustered into subgroup 2, and those with columnar fruit were mainly clustered into subgroup 4; subgroup 3 contained a mixture of fruit shapes (Figure 3).

### 2.3. GWAS of Fruit Shapes and Areca Germplasms

Among the materials tested of the 137 areca germplasms, the fruit shapes were roughly divided into spherical (fruit shape index, FSI < 1.38), oval (1.38 < FSI < 1.73) and columnar (FSI > 1.73) based on the areca FSI (Appendix A). We performed genome-wide association analysis (GWAS) by using a compressed MLM model on the data of fruit-shape traits (Appendix A). Then, we set the *p*-value of the threshold as 1.25 × 10^−3^ to detect significant loci. In the MLM association analysis of the 45,094 SNPs and areca fruit shapes, we finally obtained 200 loci that were significantly associated with fruit-shape traits. After annotating genes located at or within 50 kb of the top 200 most significant loci, 86 candidate genes were identified (Figure 4 and Appendix A).

### 2.4. Candidate Genes

We obtained 86 candidate genes associated with fruit-shape traits, each of which was located within 50 kb of a significant mutation site. We next analyzed the Gene Ontology (GO) enrichment of the 86 candidate genes. Most genes were enriched in cellular component terms. The three most enriched molecular terms were the cellular process part, metabolic process, and regulation of the biological process (Figure 5A). In the Kyoto Encyclopedia of Genes and Genomes (KEGG) enrichment analysis, the three most enriched pathways were SNARE interactions in vesicular transport and ribosome, and zeatin biosynthesis (Figure 5B; Appendix A).

Notably, one of the candidate genes on chromosome 2 encoded UDP-glucosyltransferase in the zeatin synthesis pathway. Three ribosomal protein coding genes were found near significant sites on chromosomes 2 (one gene) and 15 (two genes). Furthermore, we identified genes that encoded the biodegradation-related proteins E3 ubiquitin protein ligase, *n*-trimethyltransferase, and *CCR4-NOT* gene on chromosomes 6, 8 and 13, respectively. Three genes that encoded proteins involved in the transport of plant organic matter (syntaxin, protein transport protein SFT1, and charged multivesicular body protein 1) were identified on chromosomes 2, 10, and 13, respectively. Because areca fruit-shape traits may be regulated by many factors, the identification of these candidate genes provides important genetic resources for the analysis of fruit shape regulation mechanisms (Appendix A).

To further mine the candidate genes that regulate areca fruit shapes, we determined the tissue expression profiles of the candidate genes (Figure 6). Several genes were expressed to varying degrees in the underground roots, aerial roots, leaves, leaf veins, female and male flowers, and seeds at different developmental stages, including two genes encoding plant hormones (*Acat_8g007110*, abscisic acid-insensitive 5-like protein 2; *Acat_2g006820*, UDP-glucoronosyl and UDP-glucosyl transferase); one gene encoding a LRR receptor-like serine/threonine-protein kinase (*Acat_3g012500*, ERL1); and seven genes encoding proteins associated with signal transduction (*Acat_2g008140*, SNF1-related protein kinase catalytic; *Acat_3g012170*, lung seven transmembrane receptor; *Acat_10g014060*, CBL-interacting protein kinase; *Acat_11g020320*, casein kinase 1-like protein; *Acat_12g013930*, serine/threonine-protein kinase RIPK; *Acat_5g011970*, NPG1; *Acat_15g012780*, phosphatase 2C) (Figure 6). These may represent useful candidate genes for areca fruit-shape traits.

### 2.5. Expression Patterns of Genes Involved in Zeatin Biosynthesis Pathway

Hormones play a regulatory role in fruit development. We identified a gene (*AcUGT85A2*) involved in the areca cytokinin (CK) metabolism based on KEGG functional annotations. *AcUGT85A2* encodes cytokinin glucosyltransferase (CGT); glucosyltransferases are key regulators of cytokinin homeostasis. Notably, the UDP-glycosyltransferase UGT85A2 is involved in the glycosylation-mediated regulation of the CK level (Figure 7A). To further explore AcUGT85A2 function, we predicted protein structures using homology modeling. The results indicated that, similar to AtUGT85A2, AcUGT85A2 has a conserved domain that binds to gluconic groups (Figure 7B). We then used qRT-PCR to obtain the expression patterns of *AcUGT85A2* in columnar, oval, and spherical areca fruits. *AcUGT85A2* was significantly upregulated in columnar fruits as compared to other fruit shapes, indicating that *AcUGT85A2* may be involved in the regulation of areca fruit shape (Figure 7C).

To further investigate whether cytokinins participate in the development of fruit shape, we identified 18 genes related to the cytokinin metabolism in the areca genome. Then, the expression profiles of these genes in different fruit shapes were obtained using qRT-PCR (Figure 8A,B). We found that most genes were differentially expressed in differently shaped areca nuts. Notably, 11 of the 18 genes (e.g., *IPT-1*, *LOG-1*, and *CKX-3*) were significantly upregulated in columnar areca fruits, as compared to spherical or oval fruits. This was consistent with the expression pattern of AcUGT85A2 (Figure 7B). In addition, three genes (*CYP735A*, *CKX-6* and *LOG-9*) were significantly upregulated in spherical areca fruits, as compared to columnar or oval fruits. In general, genes tended to be upregulated in columnar and spherical areca, as compared to oval areca. The shape-related differences in the expression patterns of genes related to the zeatin metabolic pathway in areca nuts may indicate that this pathway is involved in the regulation of areca fruit shape (Figure 8B).

## 3. Discussion

In this study, we found a high diversity of areca fruit shapes, including oval, spherical, and columnar shapes. Fruit morphology is determined by the coordination of cell division and expansion, which are basic cellular processes required for the development of all plant organs [32,33]. Phytohormones are integral to the regulation of fruit development and maturation; auxin, gibberellic acid (GA), cytokinins (CKs), and others are thought to coordinate the development of fruit shape during cell division and expansion [2]. The molecular mechanisms that underlie the auxin- and GA-mediated regulation of cell division and cell expansion have made great progress. The auxin signaling and auxin–GA interaction pathways rely on Aux/IAA, ARF, and GH3 to affect fruit cell division [34]. Previous studies have shown that four auxin-/GA-responsive ARFs (SlARF6, 8, 10, and 16) redundantly restrict cell division in the tomato pericarp during early growth [33]. However, during auxin accumulation, the transcription of these four *ARF* genes was inhibited by *miR160* and *miR167*, thus promoting cell division in the early tomato pericarp [35,36]. Furthermore, at the beginning of tomato fruit cell expansion, the increased auxin and GA levels in seeds promote the expression of *GA3ox* and *GA20ox* in fruit tissues, thereby increasing the cellular concentration of bioactive GA [37]. The ability of auxin to stimulate GA biosynthesis has been demonstrated in pears (*Pyrus* spp.), where the exogenous application of 2,4-D (a synthetic auxin mimic) stimulated the expression of *GA20ox* and *GA3ox*, increased the accumulation of bioactive GA in the fruit tissues, and upregulated both cell division and expansion [38].

In addition to auxin and GA, several studies have shown that CK input can also influence cell division and expansion. For example, Zhang et al. reported the specific downregulation of CK dehydrogenase (CKX) in short bottle gourds, indicating that CKs can act in concert with or replace auxin and GA to establish fruit morphology [39]. Our results are consistent with the previous study, as most of the *CKX* genes tended to be upregulated in columnar and spherical areca as compared to oval areca, indicating that *CKX* genes may be involved in the regulation of areca fruit shape. In *Arabidopsis*, CKs directly regulate plant size. Specifically, mutations that reduce AHP activity lead to CK insensitivity and smaller siliques, while decreased *CKX* expression results in increased CK levels and larger siliques [40]. The shape-related differences in the expression patterns of genes related to the CKs synthesis pathway in areca nuts indicate that this pathway is involved in the regulation of areca fruit shape. Furthermore, *UGT85A2* encodes UDP-glycosyltransferase, which is an important regulator of CK levels in the cytokinin synthesis pathway. A previous phenotypic analysis of two *ugt85a3* mutant lines found that *UGT85A3*, a member of the *UGT85A* family, negatively regulated the number of ovules and seeds in *Arabidopsis thaliana* [41]. It has also been demonstrated that the knockdown of *CsUGT85A2* in cucumbers induced the formation of a super ovary phenotype, leading to fuller fruits [42]. Similarly, *AcUGT85A2* was significantly upregulated in columnar areca nuts as compared to fruits of other shapes. This suggested that *AcUGT85A2* may play a role in the formation of areca fruit morphology by regulating CKs, especially DZ.

Fruit shape is a polygenic quantitative genetic trait that has a complex genetic mechanism. Previous studies have performed relatively thorough genetic mappings of fruit shape in fruit vegetable crops, such as tomato, and four important genes that regulate tomato fruit shape have been successfully located and cloned, indicating that fruit shape in tomato is co-regulated by multiple genes [11,12,13,14]. In general, gene regulation patterns are polygenic and have complex quantitative genetic characteristics. In grapes, studies based on GWAS have identified some candidate genes related to fruit shape and these genes are mainly related to LRR receptor-like serine/threonine-protein kinase, transcription factors, ubiquitin ligases and plant hormones [27]. Here, we also identified several genes related to fruit-shape traits in areca. *Acat_6g004190*, which encodes RING-type E3 ubiquitin-protein ligase SiAH1, is involved in ubiquitin-mediated proteolysis. RING-type E3 is a ubiquitin ligase that determines seed size by regulating gametogenesis and cell cycle processes. Additionally, the regulation of plant seed organ morphology by RING-type E3 has been studied extensively [43,44]. *Acat_3g012500* encodes the LRR receptor-like serine/threonine-protein kinase ERECTA, which is involved in the MAPK signaling pathway in plants. Mapping-based cloning methods have identified LRR-RLK protein kinase as a possible candidate gene for the shape of peach pits [45]. Further studies of the candidate genes identified herein using GWAS are required to further clarify the molecular mechanisms that regulate nut shape in areca palms.

## 4. Materials and Methods

### 4.1. Material Collection and Phenotype Evaluation

To dissect the genetic bases of areca nut shape, we performed a GWAS including 137 areca accessions collected from areca plantations. Of these, 125 accessions originated from the Hainan Province, China, and 12 accessions originated from Vietnam. Vernier calipers were used to measure fruit length and diameter. The fruit shape index (FSI), which is defined as the ratio of fruit length to fruit diameter, and other statistics were calculated using Microsoft Excel 2021. Areca fruit shapes were distinguished, identified, and recorded based on the FSI (Appendix A).

### 4.2. DNA Preparation and Sequencing

A modified cetyltrimethylammonium bromide (CTAB) method was used to extract genomic DNA from areca leaves [46]. After quantification using 1% agarose gel electrophoresis, the working DNA solutions were diluted to 100 ng/µL and stored at −20 °C. The hyper-seq method was used to construct sequencing libraries for 137 areca DNA samples [47]. After monoclonal detection met the requirements, the HiSeq 2500 platform was used to perform paired-end 150-bp sequencing for each constructed sequencing library.

### 4.3. SNP Calling and Annotation

We tested the raw data using the FastQC software (https://github.com/s-andrews/FastQC. accessed on 20 June 2022) [48]. The clean reads were obtained using the fastp software (https://github.com/OpenGene/fastp. accessed on 20 June 2022) by filtering the raw data that passed the test. The BWA software was not only used to establish the index file of the areca reference genome (https://ftp.cngb.org/pub/CNSA/data3/CNP0000517. accessed on 20 June 2022), but was also used for the sequence comparison of the clean reads of each sample to obtain the SAM file [49,50]. Samtools was used to reorder and convert the SAM file into a BAM file, followed by another sorting process, and the final generation of the index file [51]. Picard (https://github.com/broadinstitute/picard/releases/latest. accessed on 20 June 2022) was used to mark the java environment duplicates, which were then used to create citations for the new BAM files.

The variation detection followed the best practice workflow recommended by GATK [52]. The HaplotypeCaller application of the GATK software was used to generate each sample GVCF file (GVCF format includes all variant types, including SNPs and Indel, which need to be further filtered) [52]. Finally, the GVCF was merged using CombineGVCFs to obtain the total merged GVCF file, followed by the use of GenotypeGVCFs to extract genotypes. The filtered VCF files were then obtained using VCFtools (https://vcftools.github.io/index.html. accessed on 20 June 2022) with an MAF > 0.05 (minimum allele frequency of 0.05, post removal of rare alleles) and HWE > 0.001 (removal of loci that do not satisfy the Hardy–Weinberg Equilibrium (*p* < 0.001)) [53]. The VCFtools were further used to remove the variant loci with a >50% deletion rate of all materials, to obtain the final filtered high-quality variant loci [53]. Using SnpEff software (http://pcingola.github.io/SnpEff/. accessed on 20 June 2022), we compared the sequencing reads to the reference genome to identify mutation locations (i.e., intergenic regions, untranslated regions, upstream gene regions, and downstream gene regions) and mutation types (i.e., missense, nonsense, and silent), while simultaneously annotating the mutations [54].

### 4.4. Genetic Diversity Analysis

First, we used the dnadist program by PHYLIP (http://evolution.genetics.washington.edu/phylip.html. accessed on 22 June 2022) to calculate the genetic distance matrix for all the samples and notepad++ software (https://notepad-plus.en.softonic.com/. accessed on 22 June 2022) to save the genetic distance matrix file in an appropriate format [55]. A phylogenetic tree was constructed based on this distance matrix using the neighbor-joining method and drawn, based on the spanning tree file, using the online tool iTOL (https://itol.embl.de/itol.cgi. accessed on 22 June 2022). A principal component analysis (PCA) of the SNPs detected in the *Areca catechu* populations was performed using GCTA software [56]. Finally, we used R software (https://www.r-project.org/. accessed on 22 June 2022) to draw the PCA scatter plot and to construct a population–genetic structure matrix based on the genetic composition coefficient (Q) of each sample within each subpopulation.

### 4.5. Association Analysis

We used TASSEL 5.0 (https://tassel.bitbucket.io/. accessed on 23 June 2022) to analyze the associations between areca fruit shape and these SNPs using a mixed linear model (MLM) [57]. In this analysis, the first three PCA values (eigenvectors) were used as fixed effects in the mixed model to correct for stratification. The formula used was Y = αX + βQ + μK + e, where X represents the genotype, Y represents the phenotype, Q represents the results of the PCA, K represents the indicator matrix of the individual genetic relationships, α represents the estimated fixed effect parameter, β represents the SNP effect, μ represents the predicted random individual, and e represents the random residual, such that e~(0, δ_e_^2^). We calculated the association values and identified the SNPs that were significantly associated with areca fruit-shape traits. The contribution rates of the different SNP loci to the phenotype were calculated. The significance limit of the tolerance level was set at 0.05. Finally, we used the CMplot R package (https://github.com/YinLiLin/CMplot. accessed on 23 June 2022) to draw Manhattan plots.

### 4.6. Candidate Gene Screening

We set 100 kb as the size of the flanking region (50 kb upstream to 50 kb downstream of the significantly associated SNP markers) that corresponded to the relevant SNP loci. Based on areca SNP annotations and the functional annotations of the loci, the genes on which the loci were located were considered as candidate genes. If a locus was located simultaneously upstream and downstream of other genes, the upstream and downstream genes were also considered as candidate genes. If a locus was located in the intergenic region, the upstream and downstream genes closest to the locus were considered as candidate genes.

### 4.7. Bioinformatics Analysis

To perform bioinformatic analysis of the target protein, the online tool Phyre2 (http://www.sbg.bio.ic.ac.uk/phyre2/html/page.cgi?id=help. accessed on 15 August 2022) was used to predict the tertiary structure, function, and conserved domains of UTG85A2 in *A. thaliana* and *A. catechu*, using the homologous modeling method. The protein model with the highest predictive confidence was selected, and the 3D structure and conserved domains of the enzyme protein were mapped using PyMOL (https://www.pymol.org/. accessed on 16 August 2022).

### 4.8. Real-Time Fluorescence Quantitative PCR (qRT-PCR) Verification

RNA was extracted from areca nuts with different shapes using extraction reagents (Tiangen, Beijing, China). In addition, reverse transcription was performed with the One-Step gDNA Removal and cDNA Synthesis SuperMix (Tiangen, Beijing, China). The candidate genes were analyzed using qRT-PCR. The qRT-PCR primers of the candidate genes listed in Appendix A were designed by Primer Premier 5.0 (http://www.premierbiosoft.com/primerdesign/. accessed on 24 September 2022) and β -actin was used as an internal control. The total reaction volume of qRT-PCR was 20 μL, containing 10 μL of 2 × SYBR Green Master Mix (Tiangen, China), 1 μL of the forward and reverse primers, 1 μL of the cDNA template, and 7 μL of ddH_2_O. The qRT-PCR program was as follows: 94 °C for 30 s, 40 cycles at 94 °C for 30 s, and 60 °C for 30 s. The relative gene expressions were calculated using the 2^−ΔΔCT^ method. In this experiment, each sample was represented by three biological and technical replicates.

## 5. Conclusions

To identify candidate genes associated with areca-nut shape formation, we indexed the fruit shapes of 137 areca accessions. The phylogenetic analysis of high-quality SNP markers grouped these areca accessions into four subpopulations, with close genetic relationships among the populations. A GWAS was performed to identify the significant loci related to fruit-shape traits in the germplasm resources. The proteins encoded by the screened candidate genes, which were mainly associated with biosynthesis, degradation, signal transduction, transport, and metabolic pathways, included the plant hormone glycosyltransferase, ubiquitin-protein ligase, and LRR receptor-like serine/threonine-protein kinase. Further analysis showed that *UGT85A2*, which regulates the CK level, was differentially expressed across different areca accessions. Our results help to clarify the genetic control of areca fruit-shape traits. The identification of molecular markers closely related to fruit-shape traits is of great significance for the breeding of specific shape varieties, especially those of drupes.

## Figures and Tables

**Figure 1 ijms-24-04686-f001:**
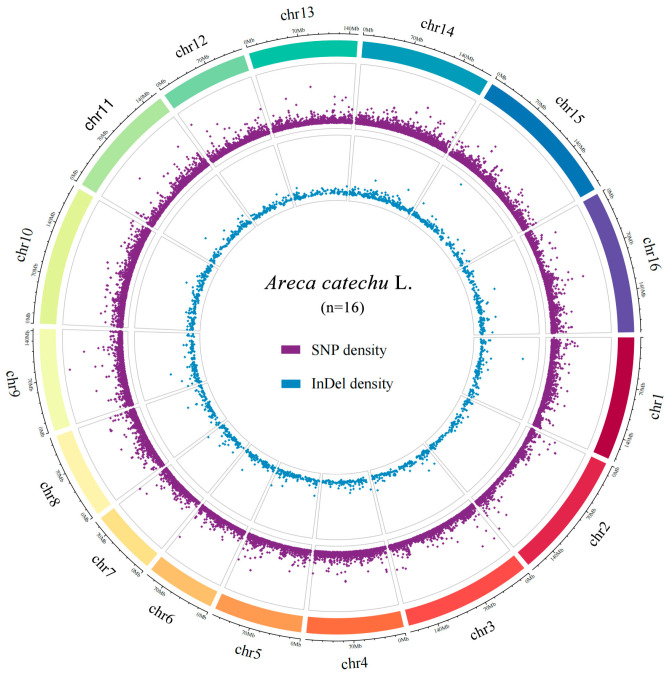
Distribution of variation detection types in areca catechu genome. SNP: single nucleotide polymorphism; InDel: insertion and deletion. The outermost circle shows the 16 chromosomes of *A*. *catechu*; purple dots in the middle circle show the distribution of SNPs across the genome; blue dots in the innermost circle show the distribution of InDels across the genome.

**Figure 2 ijms-24-04686-f002:**
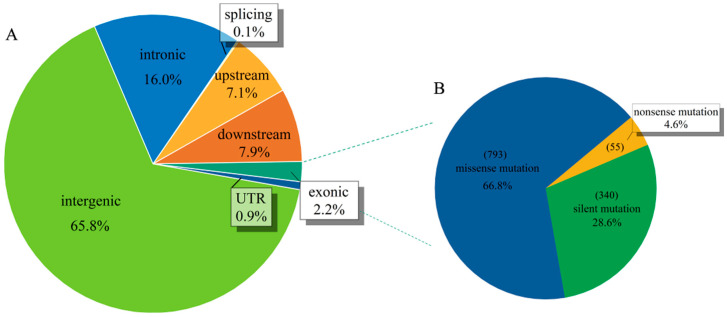
SNP location and functional annotation. (**A**) The distribution of SNP variation sites in different locations. (**B**) The relative abundance of types of SNP mutations (missense, nonsense, and silent).

**Figure 3 ijms-24-04686-f003:**
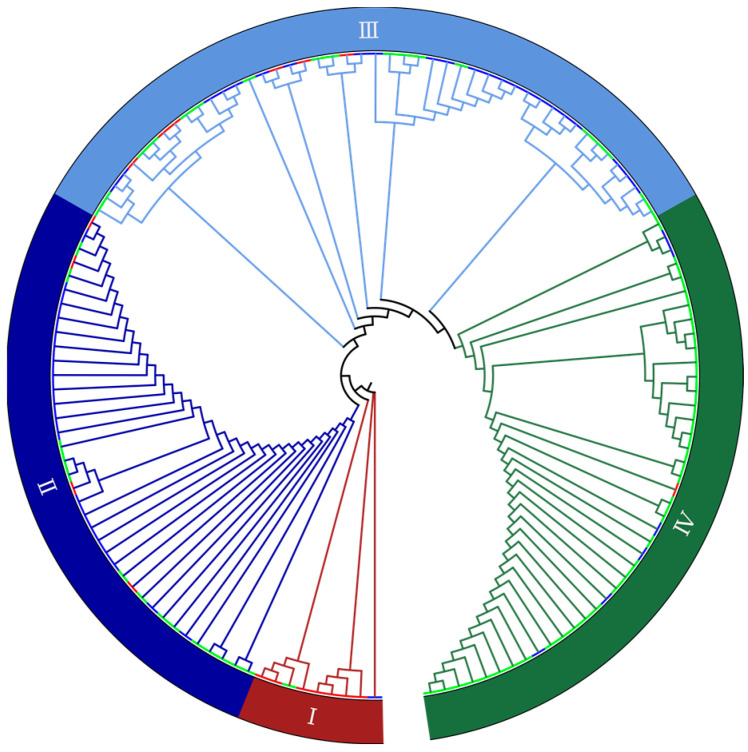
Genetic relationship analysis. Neighbor-joining phylogeny constructed based on 137 areca germplasms. Roman numbers in the outermost circle indicate subgroups. Branch color corresponds to fruit shape, with red (I), blue (II), light blue (III) and green (IV) corresponding to areca germplasms with oval, spherical, mixed shape and columnar fruits, respectively.

**Figure 4 ijms-24-04686-f004:**
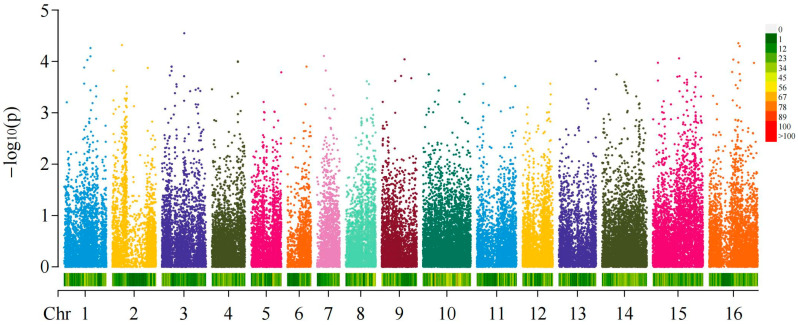
Genome-wide association analysis of the fruit shapes of areca germplasms. Manhattan plot shows the results of the whole-genome association analysis of fruit shape.

**Figure 5 ijms-24-04686-f005:**
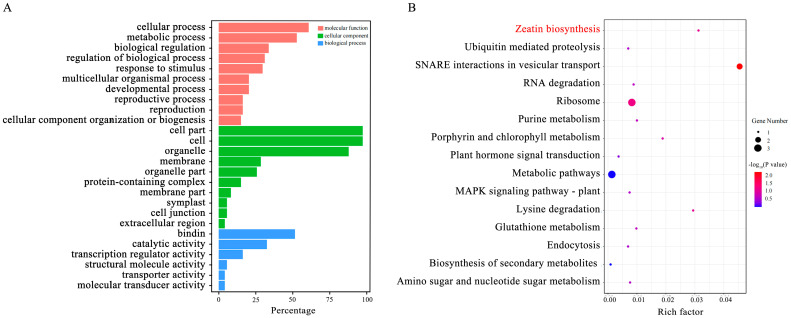
Functional annotation of candidate genes. (**A**) Gene Ontology (GO) annotation of the candidate genes. (**B**) Kyoto Encyclopedia of Genes and Genomes (KEGG) annotation of the candidate genes.

**Figure 6 ijms-24-04686-f006:**
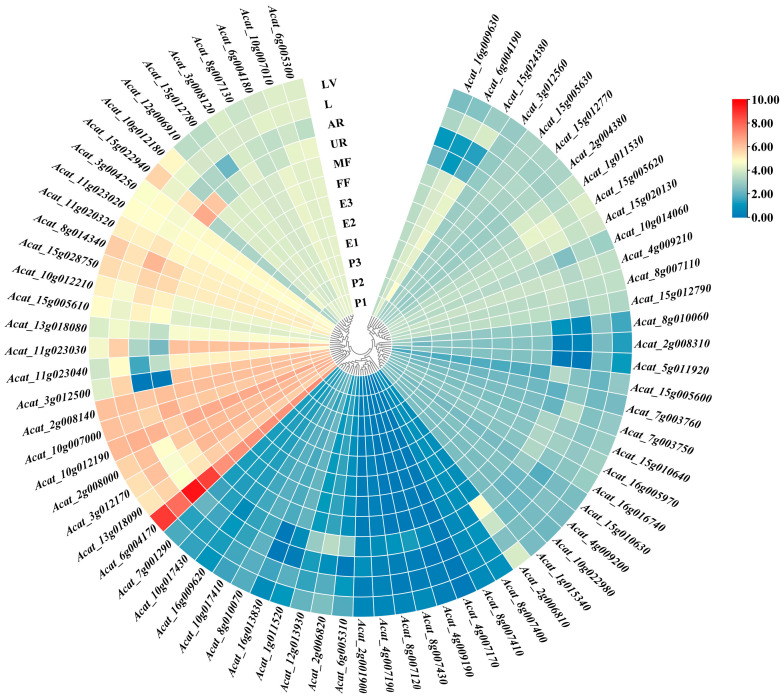
Tissue expression patterns of candidate genes annotated with areca fruit-shape traits. P1, P2, and P3: areca nut pericarp after 3, 6, and 9 months of fertilization, respectively; E1, E2, and E3: areca nut endosperm after 3, 6, and 9 months of fertilization, respectively; FF: female flower; MF: male flower; UR: underground root; AR: aerial root; L: leaf; LV: leaf vein.

**Figure 7 ijms-24-04686-f007:**
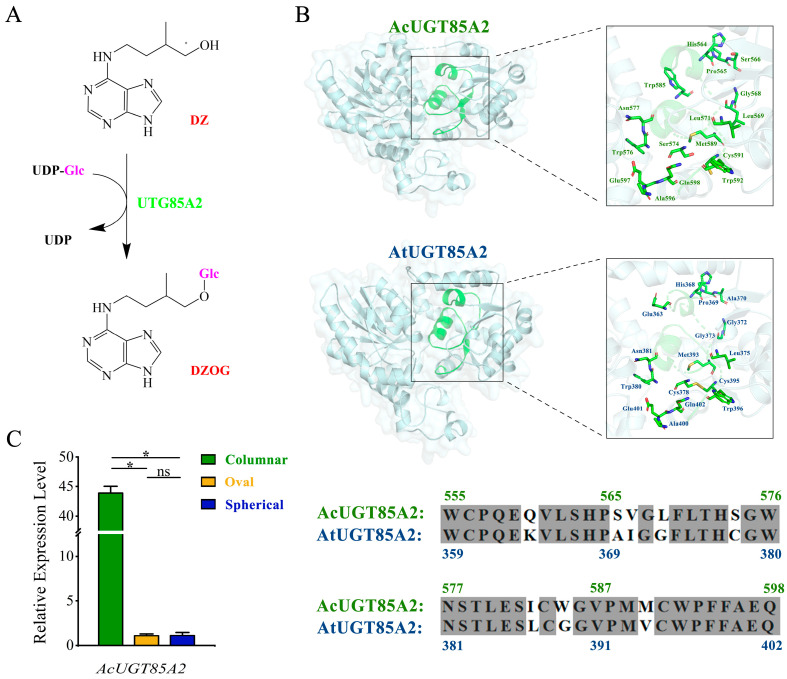
Verification of candidate genes. (**A**) Chemical structures of compounds and biosynthetic pathways related to UGT85A2. (**B**) The complete predicted structure of AcUGT85A2 (**upper left**), the functional domain of UDP-glycosyltransferase (**upper right**), the complete predicted structure of AtUGT85A2 (**lower left**), and the functional domain of UDP-glycosyltransferase (**lower right**). Differences in the amino acid sequences of the conserved functional domains of AcUGT85A2 and AtUGT85A2; fully conserved residues are shaded in gray. (**C**) Experimental verification of *AcUGT85A2* expression levels using qRT-PCR. Statistical significance was assessed by an unpaired two-tailed Student’s t-test Symbols for statistical significance levels: *: significant differences (*p* < 0.05), ns: no significant differences, n = 3.

**Figure 8 ijms-24-04686-f008:**
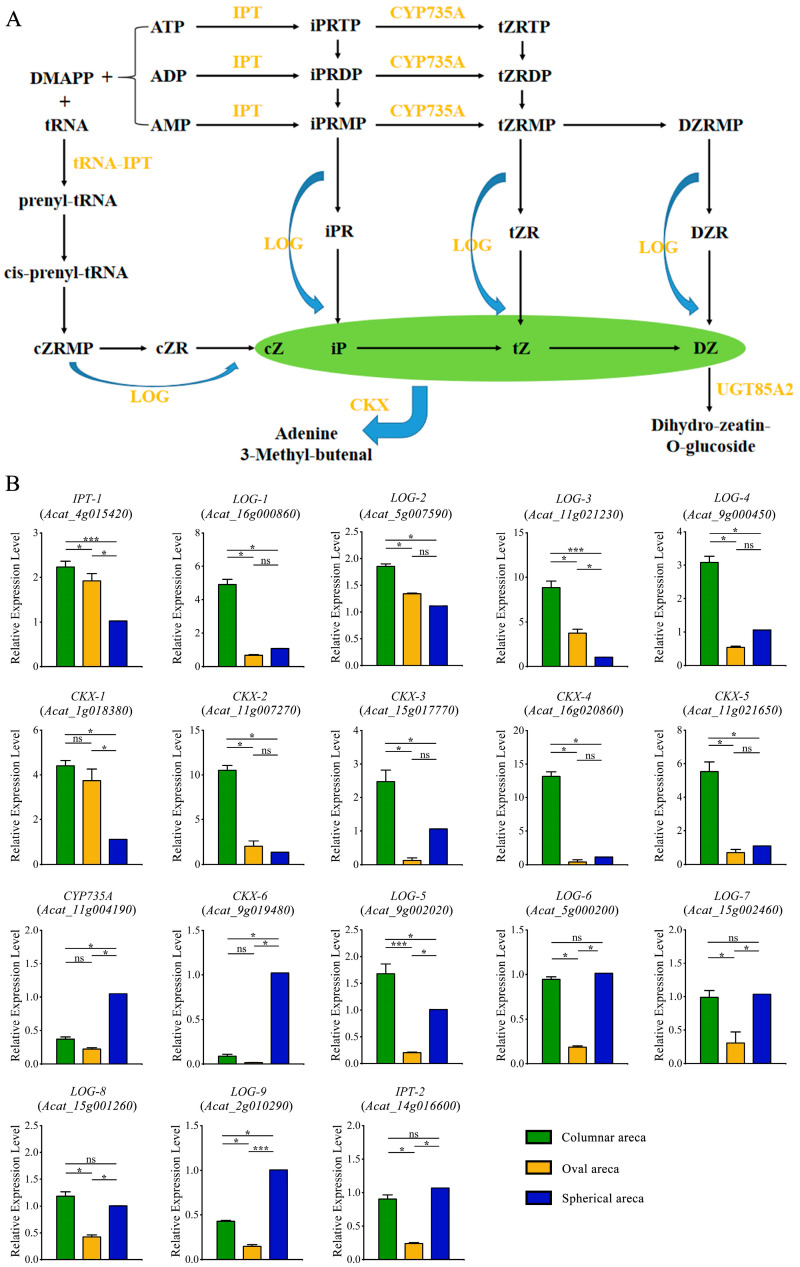
Expression patterns of genes involved in the cytokinin (CK) synthesis pathway. (**A**) The CK synthesis pathway. (**B**) Experimental verification of the expression patterns of genes associated with the CK synthesis pathway using real-time fluorescent quantitative polymerase chain reaction (qRT-PCR). Statistical significance was assessed by an unpaired two-tailed Student’s t-test Symbols for statistical significance levels: *: significant differences (*p* < 0.05), ***: extremely significant differences (*p* < 0.001), ns: no significant differences, n = 3.

## Data Availability

The original contributions presented in the study are publicly available. These data can be found at the following link: https://bigd.big.ac.cn/gvm/getProjectDetail?Project=GVM000483 (accessed on 1 February 2023).

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
