# Peer review of "Genome-Wide Association Analysis of Fruit Shape-Related Traits in Areca catechu"

_ijms, 2023, doi:10.3390/ijms24054686_

Round 1

Reviewer 1 Report

To,

The Editor,

IJMS, MDPI,

Manuscript ID: ijms-2226980

Subject: Submission of comments of the manuscript in “IJMS"

Dear Editor IJMS, MDPI,

Thank you very much for the invitation to consider a potential reviewer for the manuscript (ID: ijms-2226980). My comments responses are furnished below as per each reviewer’s comments. 

In the reviewed manuscript, the authors identified total of 45,094 high-quality single-nucleotide polymorphisms (SNPs) across the 137 areca cultivars. Phylogenetic analysis clustered the areca cultivars into four subgroups. A genome-wide association study using a mixed linear model identified the 200 loci most significantly associated with fruit-shape traits in the germplasms. In total, 86 candidate genes associated with areca fruit-shape traits were identified. Among the proteins encoded by these candidate genes were UDP-glucosyltransferase 85A2, ABA-responsive element binding factor GBF4, E3 ubiquitin-protein ligase SIAH1, and LRR receptor-like serine/threonine-protein kinase ERECTA. Quantitative real-time polymerase chain reaction (qRT-PCR) analysis showed that the gene encoding UDP-glycosyltransferase, ugt85a2, was significantly upregulated in columnar fruits as compared to spherical and oval fruits. The identification of molecular markers that are closely related to fruit-shape traits not only provides genetic data for areca breeding, but it also provides new insights into the shape formation mechanisms of drupes. In general, the manuscript represents a very big piece of information in a logical presentation. Therefore, it might be conditionally accepted subject to major revision. Authors need to address the following issues before it can be accepted for publication.

  1. I have read the entire manuscript and my initial comment is that manuscript is poorly written. I have significant concerns about the grammar and vocabulary of the manuscript; therefore, I recommend the authors to used an English proofreading service.
  2. The structure of the abstract should be improved, as well as the lack of several aspects that should be included in this section. The abstract should highlight the most important results of the parameters and characteristics assayed.
  3. The writing style of the paper is very poor. There are lots of grammatical mistakes. Long sentences with noticeable grammatical mistakes are frequently present throughout the manuscript.
  4. General note: the figures in this section are quite low resolution and difficult to make out. Higher-resolution versions will be needed for publication. Further, figure texts are not readable, for example, in Figures 1, 3, 4, 5, 6, 7B and 8B.
  5. Why you selected this crop for your experiment? Please provide the detail of the used variety.
  6. In Material and Methods:- indicate how many replicates assayed in each analysis/parameter. The number of samples or biological and technical replicates should be mentioned for each parameter in the methods.
  7. Results must be explained clearly and in detail.
  8. qRT-PCR methodology provided is also very vague and confusing. Please provide more details like what was the calibrator used in the study. I assume the authors have used the control as the calibrator. If so, the authors should not include the control within the bar graph as it represents the fold change between the treated vs control and a fold change of “1” for the ‘control’ doesn’t make any sense.  Also, would be good to provide details on what reagents (details of probes used, if any, if SYBR was used then details for that, etc.) and real time PCR machine were used in the current study.
  9. Comparison of the present results with other similar findings in the literature should be discussed in more detail. This is necessary in order to place this work together with other work in the field and to give more credibility to the present results.  
  10. Conclusion section is very lengthy. The author should emphasize this in a better way.

Reviewer 2 Report

Dear authors,

The manuscript titled “Genome-Wide Association Analysis of Fruit Shape-Related Traits in Areca catechu” provides an interesting perspective to explore the GWAS analysis in order to understand the genetic architecture of the commercial trait Fruit Shape. You found a total of 200 associated loci and 86 candidate genes, validating some genes with expression analysis via qRT-PCR. These results are very interesting in Areca breeding plans, as well as to understand the genetic basis of physiological processes of fruit formation.

In general, you have a logical sequence of analysis, in addition to having an extensive and variable panel of genotypes, as well as a high number of SNP molecular markers. However, it is strongly recommended to be more descriptive in the statistical methodology, in other words, the name of the algorithms implemented, the main parameters, and their references. The above in order to have more clarity on how the analyzes were carried out, and thus be more effective in the reproducibility of the results in subsequent works. For all the above reasons,  I recommend this manuscript for publication in IJMS with majors changes. However, I suggest to you address the following recommendations to improve some aspects before the publication (attached PDF document)

Best regards

Reviewer 3 Report

Comments to Authors:

1.      In the introduction part, lines 41 to 58 of page 2 are entirely focused on studies related to tomato fruit development, authors need to consider including a few studies related to coconut palm, date palm, or any plantation crops.

2.      In the results section I could not find the details on phenotypic measurements, which is crucial data for GWAS analysis.

3.      In section 2.3, I believe that the authors need to explain more about the GWAS results. The specified content under this section is not informative.

4.      In methods the authors mention that they have done population structure and linkage disequilibrium (LD) analysis but, I could not find the description of these results in the results section.

5.      The cited supplementary figures needed to be updated while submitting the revised manuscript.

6.      Authors also need to consider of including references to related studies to support their results.

Round 2

Reviewer 1 Report

Dear Editor,

Thank you for providing the opportunity to review the revised manuscript. The manuscript is improved considerably after revision according to the reviewer's comment. Now this study is a suitable contribution to the IJMS. I recommend the manuscript for publication.

Thank you

With best regards

Reviewer 2 Report

Dear authors,

After the corrections carry out by the authors, I recommend this manuscript for publication in IJMS in the present form.

Best regards

Reviewer 3 Report

I appreciate the authors for addressing the commenst, and modifing the manuscript. In addition, I would like to suggest the authors to go through the grammatical errors. earead came across asd came across asread came across as